# The Effect of Dietary Methyl-Donor Intake and Other Lifestyle Factors on Cancer Patients in Hungary

**DOI:** 10.3390/cancers14184432

**Published:** 2022-09-13

**Authors:** Eva Kiss, Anett Hajdu, Gertrud Forika, Magdolna Dank, Tibor Krenacs, Zsuzsanna Nemeth

**Affiliations:** 1Department of Internal Medicine and Oncology, Oncology Profile, Semmelweis University, Koranyi S. u 2/a, 1083 Budapest, Hungary; 2Department of Pathology and Experimental Cancer Research, Semmelweis University, Ulloi u. 26, 1085 Budapest, Hungary; 3Department of Internal Medicine and Oncology, Semmelweis University, Koranyi S. u 2/a, 1083 Budapest, Hungary

**Keywords:** methyl-donors, prevention, lifestyle, complementary therapy, cancer patients

## Abstract

**Simple Summary:**

The incidence of cancer has been increasing worldwide and, along with inflammation-related diseases, a significant number of cases could be prevented by appropriate lifestyle routines. Methyl-donors are macronutrients that are important in achieving a healthy balance of metabolic processes. Their deficiency can lead to several symptoms and diseases and has also been implicated in the severity of SARS-CoV-2 infection. Nutrition therapy is already recognized as an appropriate tool in the management of cancer-related fatigue to improve quality of life. In this study, we aimed to explore the potential protective effect of methyl-donor intake in breast, colorectal and pancreatic cancer by patient follow up. Our results suggest that appropriate methyl-donor intake can be useful as an adjunct of conventional oncotherapy which may contribute to improve quality of life. Whether methyl-donor intake supports cancer prevention and patient survival needs further confirmation in large patient cohorts.

**Abstract:**

Background: Nutrition is essential to life and can have an indisputable influence on health and prevention of disease development including cancer. Methyl-donors are macronutrients that are important in achieving a healthy balance of metabolic processes. Their deficiency can lead to several symptoms and diseases—even to severe SARS-CoV-2 infection. We aimed to explore the potential protective effect of methyl-donor intake in breast, colorectal and pancreatic cancer by patient follow up. Methods: A food frequency questionnaire and a diet diary were used to evaluate methyl-donor intake and blood samples were taken to evaluate Il-6 and IL-8 cytokine levels as well as *MTHFR* (C677T) polymorphism in breast, colorectal and pancreatic cancer patients. Results: We found that levels around the recommended daily intake of B6 and B9 were effective in supporting the overall survival of breast and colorectal, and a relatively higher level of pancreatic adenocarcinoma, patients. The total intake of methyl-donors significantly and negatively correlated with smoking in pancreatic cancer, while folate as well as betaine intake significantly and positively correlated with IL-8 in colorectal cancer patients. Conclusions: Our results suggest that the appropriate intake of methyl-donor can be an adjunct of conventional oncotherapy to improve quality of life. Whether methyl-donor intake supports cancer prevention and patient survival needs further confirmation in large patient cohorts.

## 1. Introduction

Nutrition is essential to life and can significantly influence health and prevent disease development [1,2]. Several papers have discussed the importance of dietary methyl-donors and related one-carbon metabolism in health and diseases [3,4]. Nutrition therapy is already recognized as an appropriate tool in the management of cancer-related fatigue to improve quality of life [5,6]. 

The incidence of cancer has been increasing worldwide; however, approximately 30–50% of cases can be prevented with appropriate lifestyle management [7,8]. Breast cancer is a heterogenous disease, which can be classified using gene and protein expression into basic subclasses based on the expression of estrogen receptor (ER), progesterone receptor (PR) and human epidermal growth factor receptor 2 (HER2). These subtypes include the ER^+^/PR^+^/HER2^+^, ER^+^/PR^+^/HER2^−^, ER^−^/PR^−^/HER2^+^ and the triple-negative ER^−^/PR^−^/HER2^−^ breast cancers, which differ in prognosis and the factors that influence their occurrence [9]. The result of epidemiological studies suggests that factors linked to hormone metabolism are more strongly related to hormone-receptor-positive breast tumors in young women, while hormone-receptor-negative tumors are more sensitive to non-hormonal factors such as nutrition [10,11,12]. Colorectal cancers are the third most common diagnoses of cancer, approximately 11% of all cancer diagnoses [13]. Colorectal cancer has a complex etiology, where both genetic and environmental factors play important roles in the development of the disease [14]. Processed meat, tobacco and alcohol consumption and obesity are considered as main risk factors for colorectal cancer development [15,16]. However, an increasing number of studies have been focusing on lifestyle factors such as dietary components, specific foods, vitamins, certain drugs or physical activity, which could reduce the colorectal cancer risk [2]. Pancreatic cancer is among the most aggressive tumor types with lower than 6 months median survival as it starts forming metastasis early [17]. Pancreatic ductal adenocarcinoma accounts for more than 85% of pancreatic cancer, which are responsible for the fifth and fourth greatest cancer-related deaths in Europe and the States, respectively [18,19]. Pancreatitis has been shown to be a risk factor for pancreatic cancer [20]. The majority of pancreatitis cases involve a combination of genetic and environmental factors; however, alcohol consumption is the best-defined risk factor [21].

Methyl-donors are food components involved in one-carbon metabolism and thus affect several metabolic processes including DNA methylation [1,3,22]. Clinical and randomized clinical trials have shown that methyl-donor micronutrient intake affects DNA methylation and thus reduces the risk of several cancer types [1]. In a systematic review and meta-analysis, we also found that the appropriate intake of B vitamins, which are also dietary methyl-donors, may reduce the risk of colorectal cancer [2]. As such, we studied the effect of methyl-donors in breast, lung and pancreatic cancer cell lines and found that methyl-donors may inhibit cancer cell growth and progression as well as inflammatory signals originating from cancer cells [23,24,25,26]. 

In our recent study, we aimed to explore the effect of dietary methyl-donor intake on survival of breast, colorectal and pancreatic cancers by patient follow up. In addition, smoking and drinking habits as well as frequency and type of physical activity were also monitored. Further, we followed laboratory parameters of systemic inflammation which are known risk factors of cancer development and progression.

## 2. Materials and Methods

### 2.1. Human Samples

We included a total of 114 cancer patients. The 41 breast cancer, 37 colorectal cancer and 36 pancreatic cancer patients (Table 1) were randomly selected in our cross-sectional study without any selection for stage, genetic, environmental or lifestyle factors, and simultaneous sample and data collections were applied in more time period (May–June 2019, March 2020, February 2021 and May–June 2021). Whole-blood samples were taken from 114 cancer patients and 9 non-cancerous controls in anticoagulated (K3-EDTA) blood collection tubes. Samples were used for DNA isolation to detect the SNP polymorphism of the *MTHFR (C677T)* gene and to measure the levels of IL-6 and IL-8 cytokines using ELISA. Our research project was approved by the Hungarian Ethical Committee of Scientific Research (No. 28123-6/2019/EÜIG).

### 2.2. Assessment of Dietary Methyl-Donor Intake, Smoking, Alcohol Consumption, Physical Activity, Marital Status and Level of Education of Selected Cancer Patients

A food frequency questionnaire (FFQ) supplemented with questions related to smoking, alcohol consumption, physical activity and level of education and a 3-day nutritional diary were used to collect information about the frequency of dietary methyl-donor consumption of cancer patients. FFQs were utilized in person, while nutritional diaries were filled out by the patients according to a previously explained and discussed guideline.

The methyl-donor intakes of the participants were calculated based on the USDA (US Department of Agriculture) Food Data Central database. Patients were stratified into low (RDA or lower), medium and high daily intake groups, as well as separated and summative methyl-donor intake. Detailed intervals are presented in Table 2. Separated intake means that daily intakes of each methyl-donor such as methionine, choline, betaine, folate and vitamins B2, B6, and B12 were calculated separately. Summative intake means that all the calculated daily intakes of the methyl-donors were pooled. 

### 2.3. DNA Isolation

Whole-blood samples were used for DNA isolation using a QIAamp DNA Blood Mini Kit (51104, Qiagen GmbH, Hilden, Germany) following the manufacturer’s protocol. Briefly, samples were thoroughly vortexed with protease and Buffer AL and incubated at 56 °C for 10 min. After that, 200 µL of 96% ethanol was added to the samples and then the mixture was transferred to special tubes with a DNA-adsorbing filter and centrifuged. Adsorbed DNAs were washed twice and finally incubated with Buffer AE solution. After centrifugation, we obtained DNAs isolated from the samples. DNA concentration (ng/µL) and quality were determined using a NanoDrop 1000 spectrophotometer (Thermo Fisher Scientific, Wilmington, DE, USA).

### 2.4. SNP Detection

We used a KASP-On-Demand kit (KBS-1016-016, LGC Genomics, Hertfordshire, UK) to determine the SNPs of the *MTHFR* (C677T) polymorphism of the patients. According to the manufacturer’s instructions, 3 µL of each DNA sample was pipetted onto 96-well PCR plates. We added a KASP genotyping mix containing 5 µL 2× KASP Master mix, 0.14 µL KASP Assay mix and 2 µL purified water per well. DNA samples were run in RT PCR according to the KASP thermal protocol determined by the manufacturer. The tested alleles were marked with FAM and HEX fluorescent dyes, where FAM marked the C allele of the *MTHFR* gene and HEX the T allele. Thermal cycles were run and data were evaluated using Applied Biosystems QuantStudio™ 3 Real-Time PCR (Thermo Fisher Scientific, Wilmington, DE, USA).

### 2.5. Quantification of IL-6 and IL-8 Cytokine Levels

Enzyme-linked immunosorbent assay (ELISA) was performed to quantify plasma concentrations of IL-6 and IL-8 cytokines using a Quantikine Human ELISA Kit (DLB50, D6050, D8000C, respectively; R&D Systems, Minneapolis, MN, USA) according to the manufacturer’s instructions. The optical density of each well was measured at 450 and 570 nm with a microplate reader (Labsystems Multiskan MS, Thermo Fisher Scientific, Waltham, MA, USA). Plasma concentrations were calculated from the standard curves and are presented in pg/mL.

### 2.6. Statistical Analysis

Our study was designed as a cross-sectional study with simultaneous single collection of data (an FFQ and a diet diary) and blood samples from patients, but the collections were divided into more time periods (see above in Section 2.1). In addition, there was a “follow-up” period from diagnosis to calculate overall survival, monitoring events of death until the end of March 2022. 

We applied descriptive statistics for the evaluation of smoking habits, alcohol consumption, physical activity, marital status and level of education. All human plasma samples were measured in duplicates. The levels of IL-6 and IL-8 cytokines were plotted as the median ± interquartile range and two-way ANOVA with Bonferroni post-tests was applied for multiple comparison. Data for survival analysis were generated in months passed from the datum of the diagnosis to death as endpoint or to the end of March 2022, when mortality data from the National Cancer Register were received. Spearman non-parametrical correlations were applied for all the correlation analyses. 

GraphPad Prism software (GraphPad Software LLC, San Diego, CA, USA) was used for all statistical evaluation except descriptive statistics, where the spreadsheet application called Numbers v11.1 (MacOS, Apple Inc., Los Altos, CA, USA) was applied. The levels of significance were determined as *: 0.01 < *p* < 0.05, **: 0.001 < *p* < 0.01 and ***: *p* < 0.001 in all cases except for ELISA, where the *p* value style of GP Prism (*: 0.01 < *p* < 0.05, **: 0.001 < *p* < 0.01, *** 0.0001 < *p* < 0.001 and ****: *p* < 0.0001) was applied. 

## 3. Results

### 3.1. Patient Characteristics

Patients characteristics are detailed in Table 1. Eighty-three females and thirty-one males from our patients were randomly selected from three groups of cancer types: 41 breast cancer, 37 colorectal cancer, 36 pancreatic cancer. All of them were treated by conventional medical treatment appropriate for the stage and progression of their diseases. The mean age was 52.95 ± 12.14 in breast, 58.68 ± 9.23 in colorectal and 60.44 ± 9.29 years in pancreatic cancer patients. The mean age of the non-cancerous control group was 33.44 ± 6.25 years. 

In the selected population, the *MTHFR C677T* polymorphism was distributed as follows: heterozygous CT—49.43%, homozygous wild CC—47.13% and homozygous mutant—3.44%. We have not found significant correlation between SNP and any of the methyl-donor intakes.

Approximately half of the patients had a higher level of education than grammar school. Most of the other half attended high school or grammar school. Only 3.57% reported elementary school as their highest level of education.

We registered the marital status of the patients as well. A total of 57.52% of the randomly selected participant were married, 15.04% were divorced, 9.73% widowed and 12.39% single.

The vast majority of the patients lived in a city/town (51.75%) and an additional 34.21% in the capital.

### 3.2. Smoking Habits, Alcohol Consumption and Physical Activity in Cancer Patients

In general, 84% of our selected cancer patients did not smoke at all; however, 14% of them smoke daily (Figure 1A). This ratio was the highest between pancreatic cancer patients, where 28% of them smoke daily, whereas 13% of colorectal patients and only 2% of breast cancer patients smoke daily (Figure 2A column). Smoking is significantly and negatively correlated with the total intake of methyl-donors (MD SUM *p* = 0.043, Spearman r = −0.350) in pancreatic cancer patients. The vast majority, 84%, of the patients did not drink alcohol more frequently than once in a month (Figure 1B). In more detail, the highest percentage of non-drinkers were found in pancreatic cancer patients, 77%. Breast cancer patients had the highest ratio of those drinking once a month or less, 47%. Data on daily drinking among pancreatic cancer patients were missing, but 3% among breast and colorectal cancer patients (Figure 2B column).

Physical activity involves both regular systematic exercises and regular daily routines, which require more muscle work than necessary for survival (Figure 1C,D). Approximately one-third of cancer patients engage in physical activity on a daily basis,74% engage in physical activity on more days a week, and only 25% engage in physical activity less often (Figure 1C,D). The frequency of physical activity shows a similar picture in all the three cancer types. However, the non-active group were missing from breast cancer patients (Figure 2C column). Investigating the type of physical activity, we found that pancreatic cancer patients mainly engaged in lightest and less specific physical activity (70%) (Figure 2D column). Breast cancer patients accounted for the highest ratio of those who engage in gymnastics, yoga and therapeutic exercises (22%), while colorectal cancer patients accounted for the highest ratio of those who engage in Nordic walking or cycling (20%) (Figure 2D column). More intensive exercises such as running or swimming have been reported less frequently in colorectal and pancreatic cancer patients (3% and 2%, respectively); however, this ratio was also quite high in breast cancer patients, 9% (Figure 2D column).

We investigated the plasma cytokine levels of IL-6 and IL-8. We found significantly higher levels of both IL-6 and IL-8 cytokines in colorectal (*p* = 0.0001 and *p* = 0.0119, respectively) and pancreatic cancer patients (*p* < 0.0001 and *p* = 0.0043, respectively) compared to controls (Figure 3). The lowest IL-6 and IL-8 cytokine levels were found in breast cancer patients and showed a significant difference compared to the other groups, although non-significant in relation to controls. The significant differences in breast cancer patients were *p* = 0.0013 (IL-6) and *p* = 0.0026 (IL-8) compared to colorectal cancer patients and *p* < 0.0001 (IL-6) and *p* = 0.0004 (IL-8) compared to pancreatic patients. In colorectal cancer patients, the level of IL-6 is significantly and positively correlated (*p* = 0.004, Spearman r = 0.479) with IL-8, and similarly the IL-8 with the intake of betaine and B9 (folate) (*p* = 0.028, Spearman r = 0.369 and *p* = 0.032, Spearman r = 0.378, respectively). Additionally, the lower level of IL-8 showed significantly longer overall survival in colorectal cancer patients (*p* = 0.032) (Figure 3). 

Applying the FFQ and 3-day diet diary, we calculated the methyl-donor intakes of each patient based on the USDA database. We calculated summative and separated methyl-donor intake, which latter related to certain types of methyl-donors such as betaine, choline, methionine, B2, B6, B9 and B12. Then, we grouped patients based on their daily intakes as low, mid and high intake (Table 2). The low group included patients in all cases if the daily intake was same as the reference daily intake (RDA) or lower. 

We examined whether different levels of methyl-donor intake could affect the overall survival of the patients, although the patient number was low. We found cancer type-dependent significant differences in the overall survival in the case of methionine, B6, B9 and the summative amounts of methyl-donor intakes (SUM MD). The low intake groups of methionine, B6 and SUM MD in colorectal cancer patients, which included the level of “RDA or lower” amounts of methyl-donors, had significantly longer overall survival (*p* = 0.014, *p* = 0.017 and *p* = 0.044, respectively) as plotted in Figure 4A,B,D. Similarly, the low B6 and B9 intake groups of breast cancer patients had longer overall survival than the other two groups (*p* = 0.012 and *p* = 0.019, respectively) (Figure 4B,C). In pancreatic cancer patients, however, the mid intake groups showed significantly higher overall survival in the case of methionine and SUM MD intakes (*p* = 0.012 and *p* = 0.015, respectively) as presented in Figure 4A,D. 

## 4. Discussion

Lifestyle factors including nutrition, smoking and drinking habits, physical activity, stress levels or socioeconomic state and behavior have a significant impact on cancer risk and development [27,28,29,30]. In our previous works, we investigated whether the intake of methyl-donor B vitamin could influence the risk of colorectal cancer [2] and also studied the methyl-donor-induced changes in the growth and apoptosis of tumor cells, and in other cellular processes [23,24,25,26]. In our recent research, we randomly selected breast, colorectal and pancreatic cancer patients and applied a food frequency questionnaire (FFQ) and a diet diary to summarize the socioeconomic status and lifestyle factors as well as evaluate whether there is an impact of methyl-donor intake on overall survival. We also analyzed the *MTHFR C677T* gene polymorphism in these cancer patients which affects the efficiency of one-carbon metabolism and therefore the DNA methylation/re-methylation pool [31]. 

Cancer is a leading cause of death worldwide, and based on research by the World Health Organization (WHO), cancer incidence is estimated to rise by 70% over the next two decades [30]. However, approximately 30–50% of cancer cases are preventable [7,8]. Dietary support is suggested as an appropriate tool both in cancer prevention [1,2] and as part of cancer therapy, i.e., in the management of cancer-related fatigue and to improve quality of life [5,6]. Dietary methyl-donors taking part in one-carbon metabolism affect several metabolic processes as well as age and metabolic state-specific DNA methylation, which is necessary for the healthy regulation of gene transcription and thus healthy metabolic regulation [1,3,22,32,33,34]. Reports and studies during on COVID-19 showed that both the very low and too high concentration of vitamins B and D, micronutrients and minerals make the human body vulnerable to chronic diseases or infections [35,36,37,38]. Indeed, we also found that the dietary intake of methyl-donors such as methionine, B6, B9 and the summative amounts of methyl-donor intakes (SUM MD) in terms of RDA levels significantly increased the overall survival of breast and colorectal cancer patients. It was only in the case of pancreatic cancer patients that a higher (mid) level of intake of methionine and summative amounts of methyl-donor intakes were found effective for significantly better survival, which could reflect the nature of this type of disease with a poorer absorption of vitamins in the small intestine. These results are also supported by our previous cell-based studies, where amounts around the RDA of methyl-donors were efficient to reduce breast, lung and pancreatic cancer cell growth and inflammation, induce apoptosis and activate related signal pathways [23,24,25,26]. Unfortunately, there was no information about the utility of the RDA intake of any dietary components similar to “drug efficacy data” (i.e., pharmacokinetic data), probably as the former are essential building blocks of the cells in the human body. Additionally, the utility of RDAs depends on several factors such as inflammation, diseases which decrease absorption, and SNPs that affect enzyme function. Moreover, to estimate the utility of methyl-donors is more complicated because the source of B vitamins for example is not only nutrition and supplementation as in the case of choline and methionine but also the gut microbiota [39].

Metabolic syndrome, including obesity with a dysregulated metabolic profile, is associated with chronic inflammation, which represents an appropriate physiological basis for cancer development [32,33,34,40]. Metabolic syndrome and subsequent cancer development are important aspects of our recent cancer research, although this link was described more than a century ago [41,42,43,44,45,46]. Moreover, the severity of inflammation could also be a predictor of progression or cancer outcome such as in the case of pancreatic cancer, where the high metastatic potential, chemoresistance and poor prognosis are closely related to the activation of nuclear factor-kB (NFkB) and give the mechanistic link to chronic inflammation in pancreatic cancer development [47]. IL-8 is a known player with pivotal roles in endothelial–mesenchymal transition (EMT) in human carcinoma cells [48]. IL-8 induces MAPK/ERK signaling and promotes lung cancer proliferation and metastasis [49] as well as colorectal cancer (CRC) cell proliferation, migration and angiogenesis, and thus is also a therapeutic target [50]. Similarly, IL-6 is elevated in both colorectal and pancreatic cancer (PC) patients and promotes proliferation and anti-apoptotic effects in CRC; further, IL-6 is correlated with advanced stages and poor survival of PC patients [51,52]. These data support our results, where significantly higher IL-8 and IL-6 levels were found both in PC and CRC patients compared to controls. Additionally, we found that CRC patients with a lower level of IL-8 had significantly better overall survival. Moreover, we found a significant and positive correlation between the IL-8 inflammatory cytokine level and methyl-donor betaine and folate (B9) intake in CRC patients, which supports the link described earlier, namely that methyl-donors in appropriate amount have anti-inflammatory properties [53]. 

Studies suggest social determinants such as lower education, lower income, and type of residency also influence the incidence of diseases including cancers. Based on a WHO survey, 56% of new cancer cases and 63% of cancer deaths occur in less developed countries [54]. Inequalities in cancer incidence and mortality exist both in relation to cancer types based on geographical appearance, with 85% of new cervical cancer cases occur in developing countries [55], and in relation to cancer types based on social level [56]. Lower social levels are connected to more cases of respiratory and stomach cancers in both genders, and in females to more cases of cervical cancers. Higher social levels in men are associated with colon and brain cancers, and with colon, breast, ovary and skin cancers in women [56]. Hungary, as a developed country, has a better health care system and a higher level of income compare to developing counties. Additionally, we found that in our study, approximately 85% of the selected patients lived in a town or a bigger city including the capital; therefore, expectedly, they had a higher social level and income compare to people living in the countryside. Approximately half of the patients have higher level of education than grammar school, which also reflects a higher social level. Thus, our results are similar to the above-mentioned international surveys, where colorectal and breast cancers are found in a higher ratio among people with a better income or social level.

Marital status is reported as a causal factor in all-cause and cause-specific mortality, and cancer survival is poorer among the unmarried population [57,58]. Marital status and mortality presumably reflect the support and care from the family members and also better financial circumstances. Moreover, these factors may be related to the tendency to avoid risky, unhealthy behavior with regard to family responsibilities and social control [59]. Thus, single males showed a decreased risk for lung cancer but an increased risk for esophageal cancer, while divorced men had an increased risk for mouth and pharynx, liver, skin and brain cancer. An increased risk of breast cancer in single females was more pronounced in older age, while an increased risk of several sites of cancer in divorced women was more prominent in younger age group [60]. Low social support was associated with a higher risk of incidence and mortality of colorectal cancer in men, which suggests that social support may affect colorectal cancer onset and prognosis via several factors including healthier lifestyles and adherence to therapeutic regimens [61]. In contrast, our study showed that more than 60% of the selected cancer patients were either married or in a relationship and only approximately 12% and 15% were single or divorced, respectively. This suggests that there are some risk factors even among married patients or those in relationships. Indeed, another study showed that marital quality and level of distress are also risk factors which are important in survival and recovery after breast cancer, i.e., in the non-distressed group, healthier dietary habits and more balanced physical activity were also described [62,63]. 

The SNP of the *MTHFR C677T* gene is an important factor in the one-carbon cycle as minor allele carriers have reduced enzyme activity with subsequent lower efficiency of the folate cycle and thus the DNA methylation pool [31]. We have not found a correlation between the SNPs and methyl-donor intake; therefore, our results related to methyl-donor intake and overall survival come from methyl-donor intake and show that these dietary components may have an impact on breast, colorectal and pancreatic cancer outcome.

An important lifestyle factor in addition to nutrition and physical activity is the smoking, where we found a significant negative correlation with total methyl-donor intake in pancreatic cancer patients. This is an additional observation which results in a lower intake and subsequent lower input of methyl-donors with the well-known association of inflammation level in smokers and worse gastrointestinal (GI) absorption of food component as a consequence of an inflamed GI tract and an altered gut microbiome [64].

A limitation of our study is the relatively low number of patients, although data were efficient for significant results. Unfortunately, the COVID-19 pandemic limited our access to more clinical datasets, as oncology outpatient treatment rules changed. This also affected some blood taken as patients arranged that with an outside institute for a faster treatment process, which resulted in the reduced number of this type of sample to measure gene polymorphism and plasma cytokine levels. A further limitation was the non-efficient anamnesis of the patients in the medical records. We have patients from the whole country and no common national medical recording system yet, only local. Therefore, outside data need to be recorded again and several times; therefore, there are missing data or data are not easy to find for stage, progression level, time, etc. This and low patient numbers did not enable a multifactorial analysis for survival analysis to eliminate confounding factors. However, because the patients were randomly selected without any background information, there is very little chance that any confounding factors would correlate with the methyl-donor intake of the patients and thus significantly bias the results of analysis related to methyl-donor intake. Additionally, we have not found any correlations between the *MTHFR C677T* SNP and methyl-donor intake which exclude the effect of reduced enzyme function with subsequently less sufficient utilization. Another limitation is the FFQ as a source of measuring the frequency of different lifestyle factors or nutrition type and the diet diary to record the daily food intake for 3 days will always have some recall bias, even though we rechecked them with the patients. 

## 5. Conclusions

In summary, our results show that appropriate methyl-donor intake can be a useful adjunct of regular onotherapies, which reduce systemic inflammation and support the quality of life of cancer patients. Methyl-donor intake may also contribute to increased overall survival of breast, colorectal and pancreatic patients, which, however, needs further confirmation in large patient cohorts. 

## Figures and Tables

**Figure 1 cancers-14-04432-f001:**
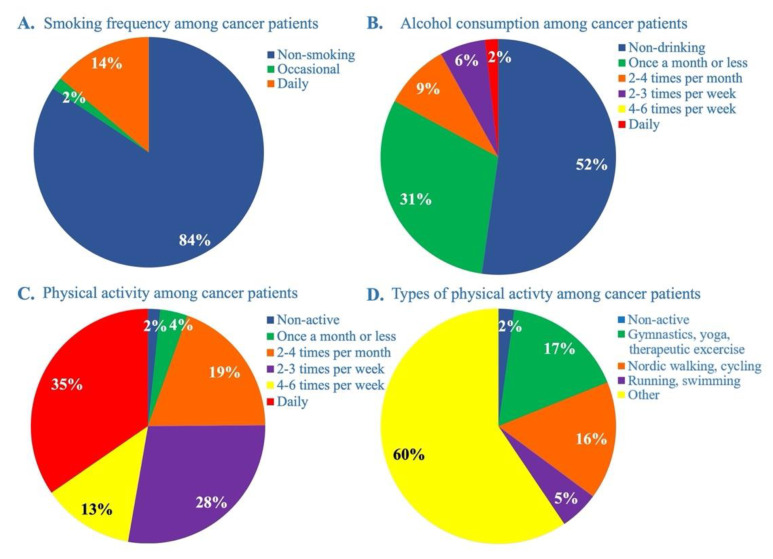
**Smoking and drinking habits, and physical activity in cancer patients.** (**A**) Frequency of smoking, (**B**) frequency of alcohol consumption, and (**C**,**D**) frequency of physical activity and types in cancer patients.

**Figure 2 cancers-14-04432-f002:**
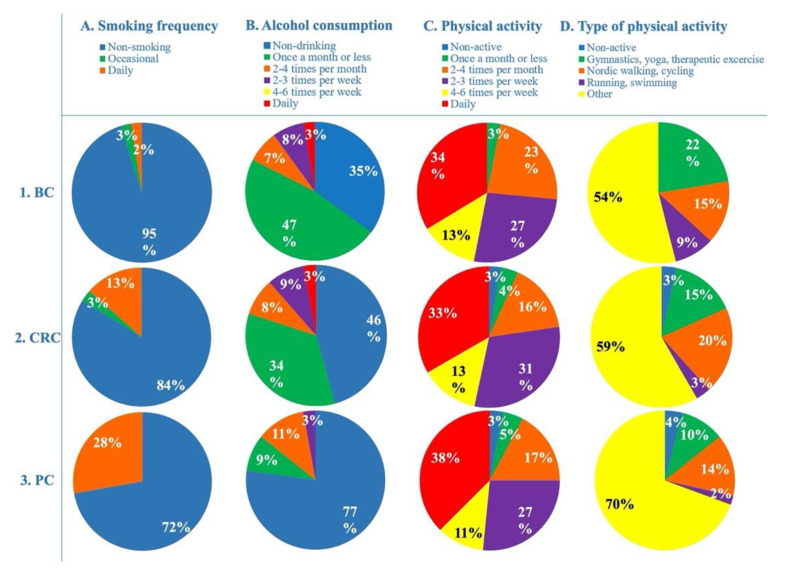
**Frequency of smoking, alcohol consumption and physical activity/type in breast. 1.** (**A**–**D**), **colorectal 2.** (**A**–**D**) **and pancreatic cancer patients 3.** (**A**–**D**). BC—breast cancer patients, CRC—colorectal cancer patients, and PC—pancreatic cancer patients.

**Figure 3 cancers-14-04432-f003:**
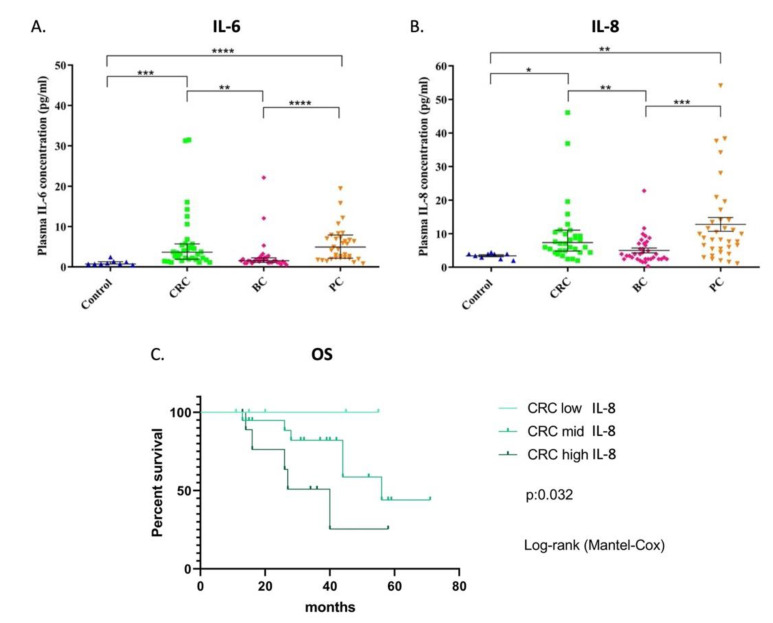
**IL-6** (**A**) **and IL-8** (**B**) **cytokine levels in colorectal, breast and pancreatic cancer patients. Overall survival in colorectal cancer patients based on the plasma level of IL-8 cytokine** (**C**)**.** *: 0.01 < *p* < 0.05, **: 0.001 < *p* < 0.01, *** 0.0001 < *p* < 0.001 and ****: *p* < 0.0001.

**Figure 4 cancers-14-04432-f004:**
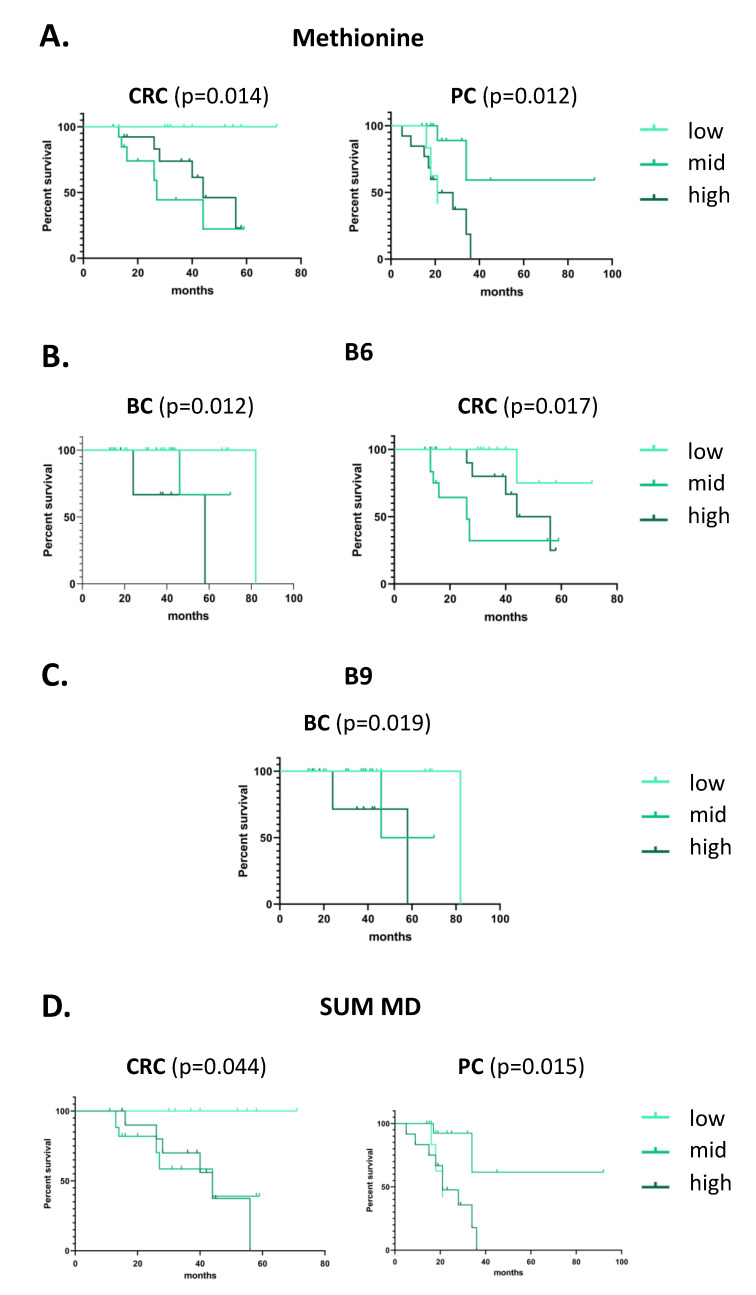
**Overall survival of the breast, colorectal and pancreatic cancer patients based on their daily intake of methyl-donors.** BC: breast cancer, CRC: colorectal cancer, PC: pancreatic cancer, B6: vitamin B6 intake, B9: vitamin B9/folate, and SUM MD: total methyl-donor intake.

**Table 1 cancers-14-04432-t001:** Characteristics of the patients with breast, colorectal and pancreatic cancer.

Characteristics	N	%
**Patients all**	114	100
female all	83	73
male all	31	27
breast cancer	41	35.96
colorectal cancer	37	32.45
pancreatic cancer	36	31.57
**Characteristics**	**Mean ± SD**	
**Age** (years, mean ± SD)		
breast cancer	52.95 ± 12.14	
colorectal cancer	58.68 ± 9.23	
pancreatic cancer	60.44 ± 9.29	
**Characteristics**	**N**	**%**
** *MTHFR C677T* **	87	100
CC	41	47.13
CT	43	49.43
TT	3	3.44
C%	62.5	71.84
T%	24.5	28.16
**Education**	112	100
postgradual	3	2.68
university/collage	46	41.07
higher vocational education	6	5.36
grammar school	29	25.89
high school	24	21.43
elementary school	4	3.57
**Marital status**	113	100
single	14	12.39
in relationship	4	3.54
registered partnership	2	1.77
married	65	57.52
divorced	17	15.04
widow	11	9.73
**Residence**		
capital city	39	34.21
county seat	2	1.75
county town	1	0.88
city	50	43.86
town	9	7.89
village	13	11.4

**Table 2 cancers-14-04432-t002:** Table for the groups of low, mid and high daily intakes of methyl-donors.

Methyl-donors listed only if significant difference were detected in relation to OS	**Intervals of mehtyl-donor intakes (mg)**		
**Types of methyl-donors**	**Low**	**Mid**	**High**
**methionine** (RDA 19 mg/kg; 1140 mg/60 kg)	≤1030.07	1030.07 < x ≤ 1503.58	1503.58<
**B6** (RDA 1.3–2 mg)	≤2.084	2.084 < x ≤ 2.572	2.572<
**B9** (RDA 0.4 mg)	≤0.399	0.399 < x ≤ 0.546	0.546<
**SUM MD**	≤1278.48	1278.48 < x ≤ 2189.79	2189.79<

## Data Availability

The data presented in this study are available in this article.

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
