# Peer review of "The Effect of Dietary Methyl-Donor Intake and Other Lifestyle Factors on Cancer Patients in Hungary"

_cancers, 2022, doi:10.3390/cancers14184432_

Round 1

Reviewer 1 Report

The relationship between lifestyle and cancer has been an interest topic. This paper mainly talked about methyl-donor intake and some of the social factors. Overall the paper raised a good investigating question and is easy to understand. However, there are some issues needs to be improved and or clarified.

1. Although dietary and social factors may play an important role in cancer prognosis, the type/stage of the disease is also a key factor. The paper reported survival data compared between Low, Mid and High methyl-donor intake patients of 3 different type of cancer. However, I didn't see a baseline analysis shows if other prognosis related factors were balanced between the 3 methyl-donor groups. Ideally, multifactor analysis should be used to included all the potential prognosis related factors. Considering the low number of patients in each group, current data is not enough to perform such analysis. Therefore, it is really important to report the baseline status between the methyl-donor groups, so that we can evaluate if there is any bias.

2. Please including more in method section. E.g. length of follow-up, endpoint of survival analysis(relapse or death), time point when blood sample was taken.

Author Response

Authors' reply to Review1 report:

Thank you for your comments and suggestions for correction of our manuscript. Please see our detailed answers and changes (in red) in the revised manuscript as listed below.

  1. "......Considering the low number of patients in each group, current data is not enough to perform such analysis. Therefore, it is really important to report the baseline status between the methyl-donor groups, so that we can evaluate if there is any bias."

Our study design was a cross sectional study where data and blood collections were made simultaneously but only one time. Also, we randomly selected the patients without any selection for stage, grade, progression of the diseases or genetic background and lifestyle. Thus, any confounding factor with could correlate to the methyl-donor intake and the related results, therefore produce concordant bias has little chance in my opinion. Additionally, methyl-donor intake did not correlate the SNP-s therefore the reduced enzymatic function, and related altered methylation, and consequences similarly can not affect the methyl-donor related survival. The low patient number could scale up the risk of bias, but the random selection decreasing these options.

We detailed this explanation in the discussion section as well in the line 428-425.

  1. "Please including more in method section. E.g. length of follow-up, endpoint of survival analysis (relapse or death), time point when blood sample was taken."

We complemented the "2.1 Human samples" (line 92-96) and the "2.6 Statistical analysis" (line 152-156 and 161-163) section with more details with the study design, endpoint of survival, time points of blood taken and data collection, and length of follow-up.

Line 92-96: " The 114 cancer patients: 41 breast cancer, 37 colorectal cancer and 36 pancreatic cancer patients (Table 1) were randomly selected in our cross-sectional study without any selection for stage, genetic, environmental or lifestyle factors, where simultaneous sample and data collections were applied in more time sections (May-June 2019, March 2020, February 2021 and May-June 2021)."

Line 152-156: " Our study was designed as a cross-sectional study with simultaneous single collection of data (FFQ and diet diary) and blood samples from patients, but the collections were divided into more time-sections (see above in 2.1.). In addition, there was a "follow-up" period from the diagnosis to calculate overall survival, monitoring events of death until the end of March 2022.

Line 161-163: "Data for survival analysis were generated in months passed from the datum of the diagnosis to death as endpoint or to the end of March 2022, when mortality data from the National Cancer Register were received."

Additionally, we added so other references and corrected spelling, grammar in some cases. All the changes labelled as red in the text.

Reviewer 2 Report

Dear Editor and Author,

The article entitled 'The effect of dietary methyl-donor intake and other lifestyle factors on cancer patients in Hungary' is a nice piece of scientific and research work. In my opinion, whole paper does not raise any substantive doubts. It is written in clear and transparent scientific language. All abbreviations are clearly explained. The methodology, results and conclusions are detailed. As a clinician, I see the results as usefull for the clinical use in future therapy. I fully recommend accepting the paper for publishing in Cancers. Regards

Author Response

Authors' reply to Reviewer 2:

Thank you for welcoming our work and the positive feedback of its scientific value, especially since you are a clinician.

We work on to ensure that our findings will be useful to clinicians and ultimately to patients, and generally to support the health of a wider and wider population.

However, we know that there is still a long way to go before this basic lifestyle information will be widely acknowledged, accepted and applied both in people's normal lives and in health services.

Round 2

Reviewer 1 Report

Appreciate the authors' response to the review comments. The issues have been addressed to a large extent.